# Application of a Fluorescent Probe for the Online Measurement of PM-Bound Reactive Oxygen Species in Chamber and Ambient Studies

**DOI:** 10.3390/s19204564

**Published:** 2019-10-21

**Authors:** Reece Brown, Svetlana Stevanovic, Zachary Brown, Mingfu Cai, Shengzhen Zhou, Wei Song, Xinming Wang, Branka Miljevic, Jun Zhao, Steven Bottle, Zoran Ristovski

**Affiliations:** 1ILAQH (International Laboratory of Air Quality and Health), Queensland University of Technology (QUT), George St. 2, Brisbane 4000 QLD, Australia; reece.a.brown@outlook.com (R.B.); svetlana.stevanovic@deakin.edu.au (S.S.); ze.brown@connect.qut.edu.au (Z.B.); b.miljevic@qut.edu.au (B.M.); 2School of Engineering, Deakin University, Geelong 3216 VIC, Australia; 3School of Atmospheric Sciences, and Guangdong Province Key Laboratory for Climate Change and Natural Disaster Studies, Sun Yat-sen University, Guangzhou 510275, China; mingfu0721@hotmail.com (M.C.); zhoushzh3@mail.sysu.edu.cn (S.Z.); zhaojun23@mail.sysu.edu.cn (J.Z.); 4Institute of Tropical and Marine Meteorology/Guangdong Provincial Key Laboratory of Regional Numerical Weather Prediction, CMA, Guangzhou 510640, China; 5State Key Laboratory of Organic Geochemistry, Guangzhou Institute of Geochemistry (GIG), Chinese Academy of Sciences, Guangzhou 510640, China; songwei@gig.ac.cn (W.S.); wangxm@gig.ac.cn (X.W.); 6School of Chemistry, Physics and Mechanical Engineering, Queensland University of Technology (QUT), Brisbane, 4000 QLD, Australia; s.bottle@qut.edu.au

**Keywords:** oxidative stress, reactive oxygen species, profluorescent nitroxide, particulate matter, aerosol, BPEAnit

## Abstract

This manuscript details the application of a profluorescent nitroxide (PFN) for the online quantification of radical concentrations on particulate matter (PM) using an improved Particle Into Nitroxide Quencher (PINQ). A miniature flow-through fluorimeter developed specifically for use with the 9,10-bis(phenylethynyl)anthracene-nitroxide (BPEAnit) probe was integrated into the PINQ, along with automated gas phase corrections through periodic high efficiency particle arrestor (HEPA) filtering. The resulting instrument is capable of unattended sampling and was operated with a minimum time resolution of 2.5 min. Details of the fluorimeter design and examples of data processing are provided, and results from a chamber study of side-stream cigarette smoke and ambient monitoring campaign in Guangzhou, China are presented. Primary cigarette smoke was shown to have both short-lived (t_1/2_ = 27 min) and long-lived (t_1/2_ = indefinite) PM-bound reactive oxygen species (ROS) components which had previously only been observed in secondary organic aerosol (SOA).

## 1. Introduction

Exposure to atmospheric particulate matter (PM) is strongly linked to increases in morbidity and mortality throughout the world [1]. Analysis of long-term exposure studies has shown a dose-dependent relationship between PM_2.5_ mass and both cardiovascular and respiratory mortality [2]. This relationship also holds for neurological disorders, with a wide scale meta-analysis also showing a link between PM_2.5_ mass exposure and a significantly increased risk of Alzheimer’s disease as well as increased risk for stroke, dementia, Parkinson’s disease and autism spectrum disorder [3]. An underlying mechanism thought to be responsible for PM to generate such a diverse range of health impacts is known as oxidative stress.

Oxidative stress is drawn from microbiology [4], and is defined as an excess of oxidants within a cell leading to disruptions in normal redox processes and cellular function. The oxidants in question are a group of oxygen-centred chemical species known as reactive oxygen species (ROS), which include: O_2_^−.^, HO^.^, RO^.^, ROO^.^, ^1^O_2_, ONOO^−^, H_2_O_2_ and ROOH. In the context of air pollution, it states that inhalation and deposition of PM in the lungs can introduce ROS to cells. Their high reactivity allows them to interfere cellular function, resulting in one of three tiers of oxidative stress depending on severity of exposure [5]. Tier 1 response results in the activation of a cells internal antioxidant defense to deplete excess ROS. If this defense is overwhelmed the cell moves into Tier 2, in which cellular signalling pathways are activated, and the affected cells become inflamed. In cases of extreme oxidative stress, Tier 3 results in the death of exposed cells through apoptosis or necrosis. These cellular responses, coupled with the ability for portions of ultrafine PM to penetrate into the bloodstream and cells [6], and the important role oxidative stress plays in cellular signalling pathways [7], forms the basis of the science implicating PM exposure to a diverse range of health outcomes.

An important parameter in the investigation of oxidative stress is the oxidative potential (OP) of PM, which is defined as the total degree to which a PM sample can oxidize components in its environment [5]. This is directly tied to how much ROS a PM sample can introduce to the body, and thus is related to the degree to which it can induce oxidative stress in exposed cells [8]. Cellular measurements of OP involve the exposure and examination of animals or lab cultivated cells for markers of oxidative stress; or direct measurement of radicals within cells using chemical probes [9]. By necessity these methods are complex and difficult to adapt to field measurements for atmospheric pollution exposure; limiting time resolution and making large-scale atmospheric studies challenging. To resolve this, several methodologies have been developed which measure the ROS activity of PM outside of cells.

Acellular methodologies vary considerably in both sensitivity and application. Notable amongst them is electron paramagnetic resonance (EPR) spectroscopy, which is uniquely capable of identifying individual ROS species using a variety of spin traps [10,11,12]. While this is a powerful capability, EPR is an offline methodology that cannot quantify all ROS using a single setup, requires extensive training, and is large, expensive and difficult to deploy in field campaigns. Alternative acellular approaches instead aim to provide a single value representing the cumulative total of ROS present independent of the species present. Systems based on these approaches [13,14,15,16] are able to collect significantly more data with a higher resolution in the field when compared to EPR. A key challenge in this approach is that ROS introduced through inhalation of PM can be divided into two major categories: 1) PM-bound ROS, defined as ROS present on particles whilst in the atmosphere [17,18]; and 2) endogenous ROS, which are ROS generated through chemical interactions between PM and exposed cells [13,19]. Both of these sources are distinct and require different methodologies for measurement. Consequently, no single acellular methodology is capable of measuring the total OP of a PM sample; instead different acellular methodologies measure contributions to total OP. This work is concerned with the most common acellular measurements of PM-bound ROS, with in-depth investigations on methodologies for both PM-bound and endogenous ROS found elsewhere [5,20].

The most commonly applied probe for PM-bound ROS is 2,7-dichlorofluorescein diacetate (DCFH-DA), a probe which reacts with certain ROS to form a fluorescent product [21]. In recent years it has been integrated into several online instruments [15,16,17,22,23,24,25]. Whilst these systems are a significant improvement over offline techniques, further advancement of the systems is made limited by the DCFH-DA probe in a few key ways, namely: a minimum reaction time of 11 min [15]; the required use of horseradish peroxidase to catalyse the reaction, which can lead to a non-linear response [21]; a high rate of autoxidation in probe stock solutions; and the requirement for the PM to be water soluble. To address these points a profluorescent nitroxide (PFN) probe 9,10-bis(phenylethynyl)anthracene-nitroxide (BPEAnit) [26] was developed and integrated into a system known as the Particle Into Nitroxide Quencher (PINQ) [27].

The PINQ is PM-bound ROS instrument which collects PM regardless of composition directly into a solution of DMSO and the BPEAnit probe. The reaction with the probe is diffusion limited, lowering the instrument time resolution to as low as one min while achieving a limit of detection of 0.08 nmol.m^−3^. This manuscript details the continued development of the Particle Into Nitroxide Quencher (PINQ) into a fully online PM-bound-ROS monitor. In particular, the development, testing and integration of a dedicated flow-through fluorimeter for rapid quantification of the BPEAnit probe is covered. Details on the design, calibration and operation methodology of the instrument are provided; and preliminary high time resolution PM-bound-ROS measurements are presented for both a chamber study and ambient monitoring campaign.

## 2. Materials and Methods

### 2.1. Online PINQ System

This manuscript details the continuation of the development of the PINQ into a continuous online system. The original offline variant has been discussed elsewhere in literature [27]. Briefly, the PINQ system collects PM directly into a solution of DMSO and the BPEAnit probe using a purpose-built steam collection device called the insoluble aerosol collector (IAC). This liquid is then de-bubbled and input into a custom flow-through fluorimeter where the BPEAnit fluorescence intensity is measured and converted into equivalent ROS concentrations. Liquid flow rates throughout the instrument are regulated using a peristaltic pump.

#### 2.1.1. Flow Switching Assembly

Previous work with the PINQ as an offline system has shown that the BPEAnit probe has a high sensitivity to gas phase ROS [28]. In order to correct for this in real time measurements of PM-bound ROS it is necessary to either account for or eliminate this contribution to the signal. Other instruments have used charcoal denuders or wetted annular denuders to remove or collect this gas phase contribution. This approach was not adopted for the PINQ due to concerns over the gas phase removal efficiency and high ultrafine particle losses [29]; additionally future characterization of this gas phase signal may further the understanding of the relationship between gas and particle phase ROS interactions [28]. In place of a denuder, the PINQ alternates between collecting filtered and unfiltered aerosol using two ball valves and a HEPA filter. A similar strategy was applied to another online PM-bound ROS instrument using the DCFH-DA probe [24]. Wide orifice ball valves were selected over a three-way valve in order to minimize particle losses inside the system. The valves, along with the spectrometer and peristaltic pump are controlled using a dedicated LabVIEW-based application, which is also responsible for logging and analysing raw data to provide preliminary real time PM-bound ROS concentrations.

#### 2.1.2. Flow-Through Fluorimeter

Initial PINQ samples were manually collected into a quartz cuvette and measured using an offline fluorimeter [30]. These measurements were time and labour intensive, highlighting the need for a flow-through setup to continuously measure the fluorescence of the sample exiting the PINQs vortex collector. Whilst commercial desktop solutions do exist, they were too large and fragile for easy integration in the instrument and limited portability. For this reason, a combination of commercially available and custom-made components were used to create a robust and significantly lower cost miniature flow-through fluorimeter.

The original offline fluorimeter combined a pulsed xenon lamp excitation source (PX-2, Ocean Optics Inc., Largo, FL, USA) with a miniature universal serial bus (USB) spectrometer (USB2000+, Ocean Optics Inc., Largo, FL, USA). Whilst effective for offline measurements [31], the xenon lamp had relatively low excitation power in the absorbance spectrum of the BPEAnit probe. This resulted in integration times of ~1 min for accurate measurements, which limited the time resolution of online measurements. To resolve this a 5 mW 450 nm laser (CPS450 Laser Module, Thorlabs Inc., Newton, NJ, USA) was selected for the flow-through fluorimeter. This provided very high probe excitation and allowed for accurate measurements with integration times in the order of milliseconds.

Two commercial flow-through cells were unsuccessfully tested before the decision was made to develop a custom solution. The first cell (583.4-F, Starna Scientific Ltd., Essex, UK) had a relatively large dead volume which led to a very slow sample response. The second cell (Fluorescence SMA Flow Cell, FIAlab Instruments Inc., Seattle, WA, USA) had a complex internal flow path which made it prone to the entrapment of bubbles, causing a scattering effect which biased results. As detailed in Section 2.1.3, the complete prevention of bubbles entering into the cell was not possible. To address these issues a custom cell was constructed in which the internal flow-path was a narrow, straight cylinder. This removed any dead volume, unnecessary surface area and flow constraints, maximizing time resolution whilst limiting the potential for bubble entrapment. The components were held together with solid mounts and internal nitrile rubber light seals. This made the final flow-through fluorimeter a small, solid unit with no fragile connections. The lack of light collimation and optics does in theory limit the lower detection limit of the system. However, this setup was capable of quantifying the smallest quantities of reacted BPEAnit relevant to the PINQ instrument, making these additional complications unnecessary for this application. A simplified cross-section showing the major components of the fluorescence cell is shown in Figure 1a.

The cell housing is a rectangular 304 grade stainless steel block with outer dimensions of 30 × 30 × 48 mm with two end caps containing threaded flat-bottom M6 microfluidic ports for sample tubing connections. The cell itself is a 20 mm long, 1 mm internal diameter, 4 mm outer diameter quartz tube. The cell endcaps are pressed into either end of the cell housing, crushing against two Teflon seals to create a liquid tight seal between the microfluidic port and the quartz cell. The laser and spectrometer are mounted perpendicular to the direction of flow, connecting to two ports bored into the centre of two perpendicular faces in the housing. Fabricated sheet metal mounts bolt these components directly to the cell, creating a single rigid unit. The internal light paths themselves are 4 mm in diameter, with their length minimized to maximize the measured response. There is a cavity opposite the laser which acts as an excess light trap, and the entire interior of the cell is optically blacked to prevent any scattering of light influencing measurements. The resulting flow-through fluorimeter shown in Figure 1b weights approximately 590 g with dimensions of 95 × 90 × 65 mm. This setup can be modified for a variety of excitation sources and spectrometers or photodiodes. For further advice on creating a similar fluorimeter, readers are encouraged to contact the corresponding author.

#### 2.1.3. Debubbler

During the collection stage the aerosol sample is collected directly into a solution of the BPEAnit probe in DMSO. The condensational growth process used to ensure a high collection efficiency for particulate matter results in the introduction of some water into this sample, typically corresponding to ~10% of the total liquid volume of the sample. This dilution can be easily corrected for in fluorescence measurements through the use of a steam dilution factor [27]. However, this dilution effect presents another problem which complicates the measurement process. When using the DMSO-based BPEAnit solution a fine mist of bubbles is created in the sample solution when the condensational growth stage is active. These bubbles create significant noise in the fluorimeter, significantly increasing the relative error in measurements in comparison to those taken from stock solutions (i.e., during fluorimeter calibrations and background measurements). This effect is not observed when the condensational growth stage is inactive, or when the collection liquid is water-based. Hence, the bubble generation has been attributed to the exothermic mixing of DMSO and water, which could lead to both the potential outgassing of gases in the collected liquids and changes in viscosity and density of the solution during the vortex collection stage leading to entrainment of some gases in the liquid flow. As both the DMSO solution and condensational growth stages cannot be excluded from the system, a debubbler was developed to remove any bubbles prior to measurement.

Initially a gravity debubbler system was employed in which the liquid was pumped through a small reservoir in which the bubbles would rise to the surface through gravity and be removed from the flow. However, the bubbles generated in this process are too small for efficient gravitational separation causing a significant portion to remain entrained in the liquid flow. Instead, a small reservoir containing a coiled stainless-steel mesh was used, with an effective internal volume of <0.2 mL. As the liquid passes through the reservoir the high surface area of the mesh traps bubbles, removing them from the liquid stream and preventing unnecessary noise in subsequent fluorescence measurements. Over time the trapped bubbles inside the reservoir will slowly coagulate into several large bubbles which will eventually escape and enter into the fluorimeter. However, the design of the flow-through cell prevents these large bubbles from becoming trapped inside the cell. Instead, the bubbles cause brief but intense scattering events which can be easily identified and removed during post analysis, which is discussed further in Section 3.2.

### 2.2. Fluorimeter Calibration

Three flow-through fluorimeters were constructed for testing, with the third using a temperature-stabilized FLAME spectrometer (Ocean Optics Inc., USA) which has the same mounting points as the original USB2000+ the cell was designed for. These three fluorimeters were calibrated using prepared concentrations of BPEAnit-Me which are continuously cycled through the fluorimeter through the use of a peristaltic pump over a 2 min period. BPEAnit-Me is a methyl trapped product of BPEAnit, wherein it has been fully reacted with methyl radicals to form a highly stable and fluorescent molecule which is similar to the various fluorescent products BPEAnit forms when reacting with radicals in a sample.

### 2.3. PINQ Data Analysis

The dedicated PINQ software continuously logs the spectrum measured by the spectrometer inside the fluorimeter. For analysis purposes the first fluorescence peak of the BPEA-nit probe at 486 nm was used. The second peak at 512 nm was also logged as the ratio of the two peaks can be used to identify faults in the setup. Herein these are referred to as the primary and secondary peaks, respectively. The height of these peaks was directly proportional to the concentration of reacted probe in the sample liquid, and hence the amount of ROS collected. Rather than logging the height of a single pixel in the centre of the fluorescence peak response, the average of several pixels centred on the peak was used. This negligibly influences the absolute value of the measurement and significantly reduces the sample noise generated through random fluctuations in individual pixels. It is also computationally simpler than fitting curves and solving the corresponding integrals to find the area under the curve. An example of the fluorescence spectrum and the averaging windows used to calculate the primary and secondary peaks is shown in Figure 2.

In order to correct for the contributions of gas phase ROS collected during measurements, the PINQ sample continuously alternates between total and gas phase measurements through the use of the flow-switching assembly. An example of the spectrums obtained for total and gas phase samples is given in Figure 2. This results in a time series which oscillates periodically as the sample is switched between gas phase and total phase measurements. The magnitude of this signal oscillation is directly proportional to the PM-bound ROS concentration in the sample aerosol. To calculate final concentrations, the first 60 s of data after each valve switch is removed as this data represents the mixing of the two phases during the PINQ response time. The remaining data is then split into total and gas phase signals, and the mean values are calculated. The gas phase signal is then interpolated to the same time base as the total phase using the spline interpolation function in Matlab; it was also corrected for background fluorescence and converted to equivalent moles of BPEAnit-Me per cubic meter of air as detailed in a previous work [27]. Finally, the PM-bound ROS concentrations are found by subtracting the gas phase from the total phase.

### 2.4. Side-Stream Cigarette Smoke Chamber Study

In order to demonstrate the response of the PINQ in a controlled environment, a cigarette was left to smoulder for 2 min before being extinguished inside a sealed 1.2 m^3^ solid chamber with internal fans to improve mixing. After a period of 10 min the PINQ was connected to the chamber outlet where it was sampled for a period of 100 min with a valve switching rate of 75 s. An inlet on the chamber was opened to allow HEPA-filtered and charcoal-denuded lab air in to replace any air removed by the sample, resulting in a continuous sample dilution with clean air.

### 2.5. Field Measurements of Background PM in Gaungzhou, China

The real-time PINQ system was first tested for ambient monitoring applications at a rooftop sampling site at the Guangzhou Institute of Geochemistry (GIG) in Guangzhou, the capital city of Guangdong Province, China. The PINQ instrument was operated over a period of two weeks from mid-October 2017 with a valve switching rate of 150 s and a PM_2.5_ impactor on its inlet to ensure comparability with supporting instrumentation. In total, 250 nM solutions of BPEAnit in DMSO were prepared every 12 h for use in the PINQ system. At these input times aerosol and liquid flow rate calibrations were also performed. The measurements were supported by a time-of-flight Aerosol Chemical Speciation Monitor (TOF-ACSM, Aerodyne Research Inc, Billerica, MA, USA), which provides information on PM_2.5_ chemical composition in real time [32]; and a BAM-1020 Continuous Particulate Monitor (Met One Instruments Inc., Grants Pass, OR, USA), which provides PM_2.5_ mass.

## 3. Results

### 3.1. Calibration Plots of the Flow-Through Fluorimeters

The calibration plots of three flow-through fluorimeters constructed using the design discussed in Section 2.1.1 are shown in Figure 3.

The two USB2000+ based spectrometers performed remarkably similar, with similar slopes and good linearity of response through all tested concentrations of BPEAnit aside from a slight overestimation at 0.5 nM measurement. The FLAME-based instrument has a lower response intensity and corresponding fitted slope, which is attributable to the different sensor the spectrometer is based on. Despite this, the fluorimeter provided a more linear response at very low concentrations, particularly for the 0.5 nM measurement.

### 3.2. Data Analysis Methodology

Analysis of the raw data collected by the PINQ Labview application was processed using a dedicated function written in Matlab for this purpose. The results of the side-stream cigarette smoke chamber measurement and key stages in the data analysis process to calculate final values are shown in Figure 4.

Figure 3a shows the raw fluorescence response along with the final data points used to calculate the average total and gas phase values in Figure 3b once data classification, filtering and trimming have been performed. The valve position data for the flow switching assembly were used to classify the data into total and gas phase samples. The data during the mixing period after the valves alternate were removed using a setup dependent time window which was determined as detailed in a previous work [27]. The other data which were removed were those points which were influenced by bubbles passing through the cell as discussed in Section 2.1.3. These were identified as the sharp peaks shown in Figure 3a which were automatically filtered from analysis through a function based on the average rate of change of fluorescence, while the cell was developed in order to prevent bubbles from becoming trapped; very occasionally they temporarily adhere to the wall of the cell. This created short periods of skewed data which were identified and removed from analysis through distortions in the measured spectrum. In the case of this study, the total sample at 5400 s and the gas phase sample at 1725 s were lost due to prolonged bubble scattering.

The averaged total and gas phase samples (*n* = 60) along with their corresponding standard error are shown in Figure 3b. The high concentration coupled with the controlled chamber environment led to very low noise and hence small error bars for each data point. The gas and total phase were interpolated to the same time base using the spline interpolation tool in Matlab, corrected for background fluorescence and converted to equivalent concentrations of BPEAnit-Me per cubic meter of air. Finally, the particle phase was calculated by subtracting the two measured phases as shown in Figure 3c.

In this example, the major source of uncertainty in the final particle phase signal was the uncertainties in the various flow rates and calibration constants. The spectrometer noise was on the order of 0.5% of the signal and contributed negligibly to the final error. Therefore, averaging over longer time periods to minimize noise does not significantly minimize the final sample accuracy. On the contrary, the longer the averaging time, the more likely it is that the measurement will be influenced by dynamically changing ROS signals and hence will become less accurate. This indicates that the best measurement strategy for the PINQ should be to minimize the averaging time of the signal and instead maximize the sample time resolution.

### 3.3. PM-Bound ROS Half-Lives

Once generated, the high reactivity of ROS causes the concentration of PM-bound ROS to decay over time through interactions with their environment. Analysis of measured decay rates in other studies indicates the presence of subsets of ROS with different half-lives, some of which can be as low as a few minutes [22]. Whilst a dedicated campaign with supporting instrumentation for chemical composition measurements is essential to fully understand the half-lives of primary ROS, some preliminary investigation can be made from the cigarette study presented in Section 3.1.

In order to investigate the half-life of ROS in the cigarette study, the data were expressed as the fraction of ROS remaining from the initial sample and corrected for known mass loss rates in the chamber. Two different models of ROS decay were investigated using the curve fitting toolbox in Matlab, as shown in Figure 5. In Figure 5a the fitted curve assumes that all ROS contained in the PM phase decay at the same rate, with the best fit corresponding to a half-life of 120 min with a 95% confidence interval CI(110,140). Whilst this model fits most of the available data points, several early and later data points are outside the 95% confidence interval. Constraining the curve to fit certain data points leads to a poorer fit elsewhere, indicating a simple exponential decay is not sufficient to explain the observed decay of PM-bound ROS. In Figure 5b a method based on several aforementioned studies is used, in which the fitted curve divides the PM-bound ROS into subsets with different half-lives, in this case two. Originally it was attempted to fit two summed exponential decay functions to model these subsets. However, it was found that the half-life of the second exponent tended towards infinity. This indicates that the lifetime of the second ROS subset is significantly longer than the experiment length and hence is not measurable with this dataset. Therefore, the model was simplified so that the second subset was simply a constant term. This resulted in a fitted curve in which the 95% CI contained all data points and resulted in a calculated short-lived ROS half-life of 27 min with a 95% CI(23,31).

In order to investigate the half-life of ROS in the cigarette study, the data were expressed as the fraction of ROS remaining from the initial sample and corrected for known mass loss rates in the chamber. Two different models of ROS decay were investigated using the curve fitting toolbox in Matlab, as shown in Figure 4. In Figure 5a the fitted curve assumes that all ROS contained in the PM phase decay at the same rate, with the best fit corresponding to a half-life of 120 min with a 95% confidence interval (CI) (110,140). Whilst this model fits most of the available data points, several early and later data points are outside the 95% confidence interval. Constraining the curve to fit certain data points leads to a poorer fit elsewhere, indicating a simple exponential decay is not sufficient to explain the observed decay of PM-bound ROS. In Figure 5b a method based on several aforementioned studies is used, in which the fitted curve divides the PM-bound ROS into subsets with different half-lives, in this case two. Originally it was attempted to fit two summed exponential decay functions to model these subsets. However, it was found that the half-life of the second exponent tended towards infinity. This indicates that the lifetime of the second ROS subset is significantly longer than the experiment length and hence is not measurable with this dataset. Therefore, the model was simplified so that the second subset was simply a constant term. This resulted in a fitted curve in which the 95% CI contained all data points and resulted in a calculated short-lived ROS half-life of 27 min with a 95% CI (23,31).

### 3.4. Application to Ambient Measurements

The online PINQ system was field tested at a sampling campaign in Guangzhou, China in 2017. Whilst the instrument was onsite for two weeks, some outages in both the PINQ and supporting instruments resulted in only periodic overlapping data for comparisons. The most significant of these was a continuous 27 h segment of data in which both the PM chemical composition and PM_2.5_ mass were also available, as shown in Figure 6. This provides a good example of the PINQs application to online ambient measurements and will be considered for analysis in this section.

Figure 6a shows a time series of both the high time resolution and hourly averages from the PINQ. The variability in the signal is considerably higher than that measured during the previously discussed chamber study. This can be partially attributed to the lower concentrations measured leading to a lower signal-to-noise ratio in the instrument. This was expected, which is why the valve switching rate of 150 s was used here as opposed to the 75 s rate used in the previous chamber study. This increases the time the sample was averaged over, improving the sensitivity of the instrument at lower concentrations. To investigate potential relationships with specific chemical components, the PM-bound ROS signal was averaged to the same 10 min time base as the TOF-ACSM data resulting in 141 data points. The Pearson’s correlation coefficients and corresponding *p*-values were calculated for this dataset and are shown in Table 1.

## 4. Discussion

### 4.1. The Online PINQ

The development of a low-cost, portable and accurate fluorimeter for the PINQ was the largest challenge in bringing the instrument online. The resulting fluorimeter is simple, robust and very cost-effective, with the single largest expense being the USB spectrometer. This could be potentially replaced with a suitable bandpass filter and photodiode detector to further lower the cost. However, the spectrometer is both sensitive and a useful tool in troubleshooting the PINQ system and identifying bubble interference; thus, this should be considered in future applications. Both the USB2000+ and FLAME-based systems performed very well, with good linear response from 0.5 to 200 nM concentrations of BPEAnit as shown in Figure 3. In future field campaigns the FLAME-based setup will be used, as the additional temperature stabilization module prevents baseline and sensitivity shifts in the spectrometer at different ambient conditions. This simplifies data analysis and improves instrument sensitivity.

The periodic filtering of the PINQ sample to correct for gas phase contributions has been used in several previous offline studies using the BPEAnit probe, and one online system utilizing DCFH-DA [24]. The use of a denuder to scrub the gas phase would result in a considerably higher time resolution (<30 s with optimized settings) and require less complex data processing. However, given the complex nature of ROS in general, the authors are not confident that there is any system which could be confidently said to suitably scrub gas-phase ROS whilst not impacting the sensitive PM-bound ROS measurements through ultrafine particle losses. An additional benefit of gas phase ROS measurements is that they provide an opportunity to investigate the total oxidative capacity of aerosols [28]. However, there measurements are currently only semi-quantitative, with an in-depth characterization of the gas phase collection efficiency needed to fully quantify gas phase ROS measurements.

The time resolution of the originally published PINQ system was reported as one minute, which combined a 40 s mixing time with a 20 s sample averaging time [27]. This mixing time was determined by a combination of the internal volume of the flow-path and the liquid flow rate of the sample. The setup for the newer debubbler detailed in Section 2.1.3 marginally increased this internal liquid volume. This resulted in a valve switching rate of 75 s for the chamber study, incorporating a 60 s mixing time and 15 s averaging time. The necessity for measurements of both total and gas phase measurements to measure PM-bound ROS doubled the final time resolution to 2.5 min. This short averaging time is possible in chamber studies as the high concentrations cause the signal-to-noise ratio to become sufficiently high such that minimizing the averaging time will have a negligible impact on the final result. This is evident by the almost imperceptibly small error bars shown in the averaged signal in Figure 4b. The majority of the error in Figure 4c is instead the result of the collective uncertainties in the calibrations of the fluorimeter, aerosol and liquid flow rates. The switching rate was increased further to 150 s in ambient measurements, extending the averaging time to 90 s in an effort to reduce uncertainty from noise at lower concentrations.

### 4.2. Implications of the Initial Ambient Study

The 27 h of ambient air measurements presented here were not of a sufficient time length to make any conclusive comments about the chemistry of PM-bound ROS. Instead, these data are presented as an example of real-time ambient applications in comparison to that of the cigarette chamber study. As mentioned in the previous section, the averaging time in ambient measurements was significantly extended in order to reduce noise in measurements. Despite this, Figure 6a shows a high variability between individual data points. While this is certainly in part attributed to the lower signal-to-noise ratio, it is not ruled out that there is a real high short-term variability in PM-bound ROS concentrations. Potential causes of this variability could be nearby sources of highly short-lived ROS species, or unmixed plumes from nearby traffic. In order to explore this further, a better understanding of the underlying chemistry of atmospheric PM-bound ROS and its sources is needed.

The correlations presented in Table 1 indicate the PINQ measurements correlate to varying degrees with all the major ion masses measured by the TOF-ACSM. This is not an expected result and is likely due to the pollution event observed at approximately 06:00 on December 12 (see Figure 6). This resulted in increases in the PM-bound ROS, total PM_2.5_ mass and to varying degrees all mass components measured by the ACSM. PM-bound ROS and total PM_2.5_ mass both experienced a two-fold increase over this period, resulting in relatively constant ROS per PM_2.5_ mass. As such, no significant observations regarding the influence on chemical compositional on PM-bound ROS can be made over a short period. Instead, the correlations shown in Table 1 are likely due to their shared relationship with total PM_2.5_ mass. This highlights the need for significantly longer periods of data collection with systems like the PINQ if the sources of PM-bound ROS in the atmosphere are to be understood.

### 4.3. Half-Lives of PM-Bound ROS

The PM emitted during cigarette combustion has previously been shown to contain high concentrations of ROS using the BPEAnit probe [31,33]. It has also been observed that inside a sealed chamber the concentration of ROS per mass decayed over time [31]. Similar observations have been made using the DCFH probe with differing methodologies and PM sources [15,22]. This has led to the concept of ROS half-lives; wherein the high reactivity of the radicals present causes them to dissipate over time though chemical interactions with their surroundings. The measurement of these half-lives has resulted in two broad categories of ROS: short-lived ROS, with half-lives as low as a few minutes [22]; and long-lived ROS, which can exist from several hours to effectively indefinitely [15]. The analysis of the cigarette smoke chamber study presented in Section 3.2 further reinforces these findings, with two distinct half-lives present.

The first fit, which assumed all ROS exponentially decay at the same rate, was relatively poor, with fitting attempts failing to intercept all data points. In contrast to this, the assumption of two distinct subsets provides a good fit with all observed data. The long-lived term is represented as a constant, as fitting the half-life for this value generated a confidence interval several orders of magnitude higher than the fitted value. This does not indicate that this subset survives indefinitely, only that the half-life is significantly longer than the experiment length of 100 min and hence cannot be accurately determined.

The shorter-lived ROS in this study have a measured half-life of 27 min with a 95% confidence interval of four min. Whilst the use of “short-lived’ is correct in the context of this study, the half-life of these ROS is an order of magnitude longer than the estimated few minutes measured for oxidized oleic acid aerosol using a DCFH-based online system [22]. This is attributable to the large difference in PM investigated. The ozonolysis of oleic acid leads to relatively homogenous liquid droplets containing large concentrations of peroxide groups [34], which will be the dominant ROS present. In contrast, PM from cigarette smoke is a complex heterogeneous mixture of solid and liquid phase chemical species in which the majority of ROS are likely to be alkoxy radicals [12]. As ROS decay occurs through reactions with surrounding molecules, a difference in chemistry will inevitably lead to differences in ROS half-lives. However, this finding does not offer insight into the specific factors determining them. This would require an in-depth investigation using the PINQ in conjunction with multiple sources and a wide suite of supporting instrumentation providing the chemical composition of both the gas and particle phase.

### 4.4. What Time Resolution is Necessary for the Measurement of PM-Bound ROS?

An often-asked question is: what time resolution of PM-bound ROS measurements is truly necessary? As can be seen in the ambient measurements shown in Figure 6a, the high time resolution data is significantly variable and thus little can be drawn about exposure and potential health outcomes at this level. Instead, the hourly averages are more suited to this purpose. However, higher time resolutions are invaluable in examining the chemistry which drives the concentrations of PM-bound ROS. In the case of chamber studies, the PINQ provides significantly more data points, resulting in more accurate half-life calculations and the ability to directly observe the decay of short-lived species. In atmospheric measurements it provides more statistical power for correlations with other instrumentation and faster responses to pollution events. High time resolution data is not by definition a necessary requirement for atmospheric studies, but it is a valuable tool to further understanding PM-bound ROS in the atmosphere.

## Figures and Tables

**Figure 1 sensors-19-04564-f001:**
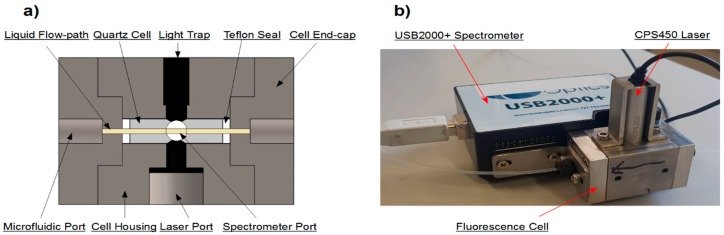
(**a**) Simplified cross-section of the microfluidic cell showing the assembly of key components. The cells outer dimensions are 30 × 30 × 48 mm. Notably, the liquid flow-path is a continuous cylinder throughout the entire illuminated path, significantly reducing the entrapment of bubbles inside the cell. (**b**) The assembly of the full fluorimeter illustrating the connections between the fluorescence cell, CPS450 laser and USB2000+ spectrometer.

**Figure 2 sensors-19-04564-f002:**
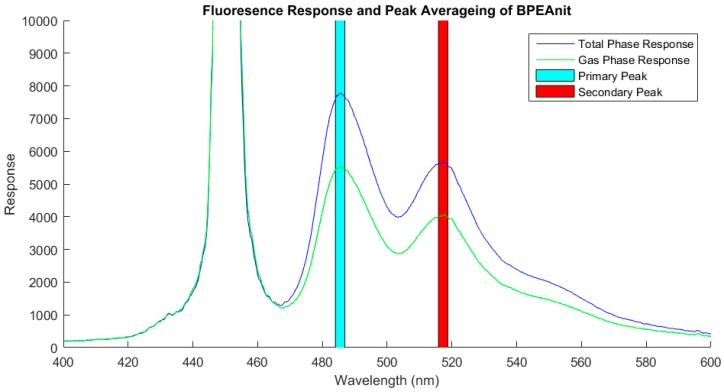
The typical fluorescence response measured for the 9,10-bis(phenylethynyl)anthracene-nitroxide (BPEAnit) probe with the flow-through fluorimeter for a total phase and gas phase sample. The response wavelengths averaged for the measurement of the primary and secondary peaks are indicated by the blue and red shaded regions, respectively. The large third peak centred on 450 nm is the laser used as the excitation source.

**Figure 3 sensors-19-04564-f003:**
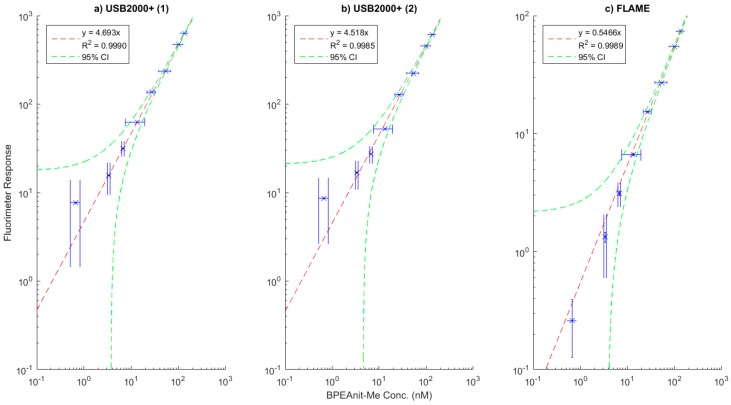
(**a–c**) The calibration plots for the three flow-through fluorimeters constructed and tested. (**a**) and (**b**) use USB2000+ series spectrometers, whilst (**c**) uses a newer FLAME spectrometer with temperature stabilization. The flame spectrometer is more adept at measuring very low concentrations, although these are far below those measured when integrated into the Particle Into Nitroxide Quencher (PINQ).

**Figure 4 sensors-19-04564-f004:**
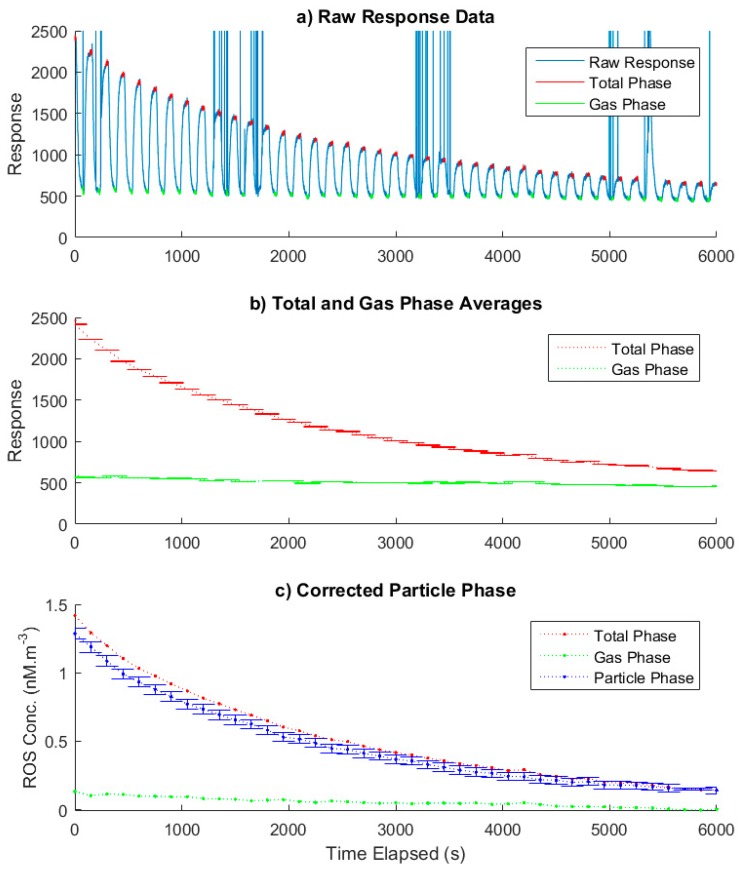
(**a**) The raw fluorescence response of the PINQ sample over time for side-stream cigarette smoke in a chamber. The alternating signal is caused by switching between filtered and unfiltered air. Sharp spikes in the signal are caused by bubbles in the sample line. The total and gas phase data points coloured indicate the data points averaged to generate the next plot in the figure. (**b**) The de-bubbled, trimmed and averaged plateaus and corresponding standard error of the alternating signal which correspond to the total and gas phase. (**c**) The final particle phase reactive oxygen species (ROS) concentration with standard error calculated by subtracting the interpolated gas phase from the total phase after correcting for background fluorescence and converting the signal to equivalent concentrations of BPEAnit-Me per cubic meter of air.

**Figure 5 sensors-19-04564-f005:**
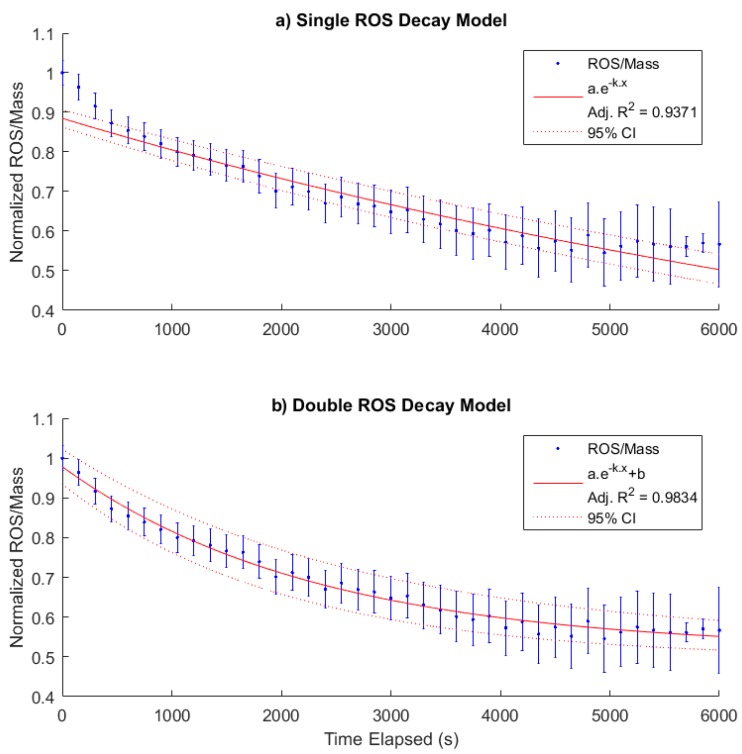
The ROS data, normalized by its initial value and corrected for mass losses, along with two different fitted curves to the ROS data using different assumptions of ROS decay. (**a**) shows a fit which assumes that all ROS present have a single half-life, with a value calculated from the fit of 120 min with a 95% confidence interval (CI)(110,140). This model does not fit well at low or high elapsed times. (**b**) shows a fit which assumes there are two subsets of ROS, in which the second set has a lifetime which is effectively infinite over the measurement period. The half-life of the short-lived ROS using this model is 27 min with a 95% CI(23,31).

**Figure 6 sensors-19-04564-f006:**
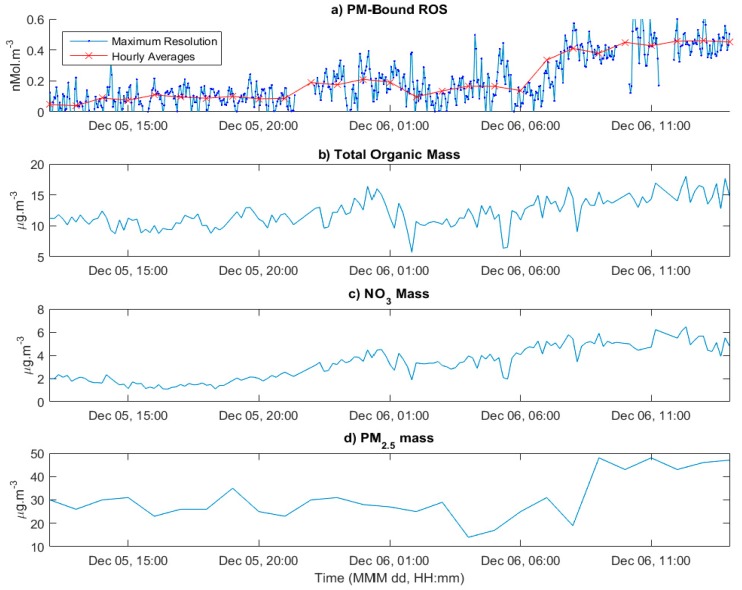
(**a**) shows a day period of analysed PINQ data collected at a rooftop site measuring ambient background aerosol in Guangzhou, China. (**b**) shows the hourly average organic mass concentration measured by the time-of-flight Aerosol Chemical Speciation Monitor (TOF-ACSM). (**c**) gives the NO_3_ mass measured by the TOF-ACSM. (**d**) shows the hourly total particulate matter (PM)_2.5_ mass concentration measured by the BAM-1020.

**Table 1 sensors-19-04564-t001:** Pearson’s correlation coefficients and corresponding *p*-values between PM-bound ROS and major TOF-ACSM fragments over the shown measurement period (*n* = 141), being chloride (Cl^−^), nitrate (NO_3_^−^), sulphate (SO_4_^2−^), ammonium (NH_4_^+^) and total organics (Org). PM-bound ROS was found to correlate best with nitrate.

	Cl^−^	NO_3_^−^	SO_4_^2−^	NH_4_^+^	Org
**Corr. Fac.**	0.39	0.65	0.28	0.50	0.54
***p*** **-Value**	8 × 10^–7^	2 × 10^–19^	6 × 10^–4^	7 × 10^–11^	5 × 10^–13^

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
