# Peer review of "Application of a Fluorescent Probe for the Online Measurement of PM-Bound Reactive Oxygen Species in Chamber and Ambient Studies"

_sensors, 2019, doi:10.3390/s19204564_

Round 1

Reviewer 1 Report

The authors described and provided information on an online OP measurement using PFN and PINQ. OP of PM is of high interest nowadays as it links adverse health effects and PM exposure in an additional, promising way. So a reliable online OP sensor is what we need. Due to this the manuscript is clearly of interest for sensors.

My main concern on the manuscript are due to the overall presentation of the thematic and also the data. This manuscript should gives a clear information why OP is relevant and what exactly is measured (at least briefly)? It should also provides easily all information needed to reproduce the sensor and the measurements. Unfortunately this is a bit missing. So I encourage the authors to work on this before possible publication.

Line 44: Please also refer to the three tiers of oxidative stress paradigm and clearly define Oxidative Potential and Oxidative Stress.

Line 58: Please provide information on further methods able to measure OP (e.g. Hellack et al. 2017 Environmental Science Nano). Provide pros and cons and especially describe the pros for a small sensor please! Discuss later on also please uncertainties, cross interferences of the sensors compared to cost intensive robust methods/instruments!

Line 100-131: Please provide the data of all these experiments (different wavelength etc.) 

Line 130-136: Please shorten this section. This information is not really needed.

Figure 1: Please add a scale

Line 154: It is not a diagram, a scheme or construction figure maybe? Also nice would be a real picture.

Line173-175: Please provide data, how you checked the effectiveness of the debubbler? No bubbles are observed means what? Not visible and if so is this an appropriate control? Please describe and explain.

Line 190-191: Please provide data (show the scattering please)

Line 209-212: Please discuss briefly why not using the integral of the area under the peak?

Line 246: Please explain if the AMS is really sampling the same PM sizes and amount like your sampler – so is it comparable? What about interferences of the sensor and mixture of PM and gases?

Line 267: Please check the figure numbers

Line 268: Please explain how it was processed in Matlab?

Line 294: Please provide for all measurements also an n and the variations.

Line 308-339: Please discuss not only half-Lives in more detail but also please ROS cycling, e.g. superoxide becomes hydroxyl radical etc. so what happened at a later time point is that you might measure the same now transformed radicals (with shorter half-time) again or not? Please also discuss in more detail what ROS species might be measured?

Line 363-366: Please provide information of n, statistic information on SD etc. Hard to believe p values < 0.05 and weak correl of e.g. 0.3 (SO4). What about partial correlation and also interferences, did the authors check this? Would be nice.

Line370: What is meant by viable?

Line 381: As briefly done please discuss if a higher time resolution is really needed especially when we want to link it to health effects?

Line419: Please see comment before and keep in mind if AMS particle sampling is comparable.

Line449: What does this mean for regulatory purpose? Please provide also briefly future aims and scenarios using OP sensors.

Reviewer 2 Report

In this manuscript, the authors reported a fluorescence probe-based online monitoring system for the measurement of PM-bound reactive oxygen species in chamber and ambient. This work is interesting and useful. And the draft is well written and organized. So, in my opinion, I recommend its acceptance. Detailed comments are listed as below.

A control of bare PM that not bound ROS should be inspected. Bare PM can be obtained by a  pre-treat procedure with ROS scavengers like ascorbic acidin presence. What is the limitation of this on-line fluorescence measurement system for PM-ROS?

Round 2

Reviewer 1 Report

The authors have thoroughly worked on the manuscript, thanks for that! I have no further comments.

Reviewer 2 Report

The questions have been addressed by the authors. I recommend its acceptance in its current version.